# Dietary Inflammatory Index and Dietary Diversity Score Associated with Sarcopenia and Its Components: Findings from a Nationwide Cross-Sectional Study

**DOI:** 10.3390/nu16071038

**Published:** 2024-04-02

**Authors:** Guzhengyue Zheng, Hui Xia, Zhihan Lai, Hui Shi, Junguo Zhang, Chongjian Wang, Fei Tian, Hualiang Lin

**Affiliations:** 1Department of Epidemiology, School of Public Health, Sun Yat-sen University, No. 74, 2nd Yat-sen Road, Yuexiu District, Guangzhou 510080, China; zhenggzhy@mail2.sysu.edu.cn (G.Z.); laizhh3@mail2.sysu.edu.cn (Z.L.); shih39@mail2.sysu.edu.cn (H.S.); zhangjg37@mail.sysu.edu.cn (J.Z.); tianf8@mail2.sysu.edu.cn (F.T.); 2Center for Health Care, Longhua District, Shenzhen 518109, China; huihui_xia521@163.com; 3Department of Epidemiology and Biostatistics, College of Public Health, Zhengzhou University, Zhengzhou 450001, China; tjwcj2005@126.com

**Keywords:** energy-adjusted dietary inflammatory index, dietary diversity score, sarcopenia, muscle strength, physical performance, muscle mass, joint association

## Abstract

Little is known about the independent and joint effects of the energy-adjusted dietary inflammatory index (E-DII) and dietary diversity score (DDS) on sarcopenia and its components (low muscle mass, low muscle strength, and low physical performance). A total of 155,669 UK Biobank participants with ≥1 (maximum 5) 24 h dietary assessments were included in this cross-sectional analysis. We used logistic regression models to investigate the associations of E-DII and DDS with sarcopenia and its three components. We further examined the joint effects of E-DII and DDS on sarcopenia and its components using additive and multiplicative interaction analyses. We observed that lower E-DII and higher DDS were associated with lower odds of sarcopenia and its components. There were significant joint associations of E-DII and DDS with sarcopenia and low physical performance (*p*-interaction < 0.05) on the multiplicative interactive scale. Our study suggests that lower dietary inflammatory potential and higher dietary diversity might be important protective factors against sarcopenia and its components. More cases of sarcopenia and low physical performance might be preventable by adherence to a more anti-inflammatory diet combined with a higher dietary diversity.

## 1. Introduction

Characterized by a decrease in muscle mass, strength, and physical performance [1], sarcopenia is a common health issue among the elderly population [2]. A recent community-based survey conducted in the UK showed that the prevalence of sarcopenia in people over 60 years of age was 7.4% among males and 11.0% among females [3]. Sarcopenia may result in high-risk mortality, physical disability, loss of quality of life, and substantial expenditure on healthcare [1,4,5]. Considering sarcopenia’s high prevalence and harmful health effects, searching for protective factors has important public health implications.

Currently, there is no effective pharmaceutical approach to treating sarcopenia [1], and dietary intervention is recognized as one of the main preventive options [6]. Previous studies have reported that reducing dietary inflammatory potential and consuming diverse foods are extraordinarily important in stopping the onset of various diseases [7,8]. Several population-based studies have analyzed the relationship between the dietary inflammatory index (DII) and sarcopenia [9,10]. However, these studies are limited by the absence of energy adjustment [11] or the use of high-risk populations only [9], hindering the generalizability of the findings.

The dietary diversity score (DDS) is an indicator that measures the overall nutritional adequacy and balance of the foods consumed by an individual [12,13]. Several studies have explored the relationship between DDS and varied health outcomes [14,15]. However, to our knowledge, no study has evaluated the associations of DDS with sarcopenia and its components. In addition, based on the common mechanisms of anti-inflammatory diet and diverse diet for preventing sarcopenia, such as anti-inflammation, antioxidative stress, and enhancement of immune function [16], it is thus reasonable to assume that following a higher dietary diversity might enhance the protective effect of an anti-inflammatory diet against the occurrence of sarcopenia. However, no study has investigated the combined associations of E-DII and DDS with sarcopenia and its components. In addition, previous studies on the relationship between diet and sarcopenia did not define sarcopenia based on the European Working Group on Sarcopenia in Older People 2019 (EWGSOP2) classification and cut-off points [17].

To fill these research gaps, we implemented a cross-sectional study of the UK Biobank data to investigate the independent and joint associations of an energy-adjusted dietary inflammatory index (E-DII) and DDS with sarcopenia and its components based on EWGSOP2.

## 2. Methods

### 2.1. Study Population

Cross-sectional survey data from the UK Biobank were used in our analyses. The UK Biobank recruited more than 500,000 adults aged 37–73 years during 2006–2010 and obtained information on socioeconomic and lifestyle factors, diet, disease status, and medication history from 22 assessment centers via physical and biological measurements, verbal interviews, and questionnaires [18].

For our analyses, we started by including 211,025 participants with ≥1 (maximum 5) 24 h dietary survey. We then excluded participants (*n* = 55,356) who met the following criteria: whose 24 h dietary recall information was unbelievable or missing (*n* = 5888), who had missing data for components of sarcopenia (*n* = 4185), who had excessively daily low caloric intakes (<800 kcal for males and <600 kcal for females) or high intakes (>4200 kcal for males and >3500 kcal for females) (*n* = 1783), or who had missing data for any of the key covariates, including sex, age, race, household income, physical activity, alcohol consumption, smoking status, BMI, and total energy intake (*n* = 43,500) (Appendix A). The characteristics of the samples before exclusion (*n* = 211,025) and the final analytic sample (*n* = 155,669) are comparable (Appendix A).

### 2.2. Assessment of Sarcopenia and Its Components

The information used to define sarcopenia was obtained through physical measurements taken. Two physical measurements associated with sarcopenia were conducted during 2006–2013. We calculated the mean of the repeated physical assessments associated with sarcopenia. This study determined sarcopenia according to the European Working Group on Sarcopenia in Older People 2019 (EWGSOP2) classification and cut-off points [17]. Three components were used to identify sarcopenia patients: low muscle mass, low muscle strength, and low physical performance. In this study, we used relative appendicular lean mass (RALM), hand grip strength, and walking pace to estimate the three elements of sarcopenia: participants with low RALM were defined as having low muscle mass, those with low hand grip strength were considered to have low muscle strength, and those with slow walking pace were considered to have low physical performance (Appendix A). Following the EWGSOP2’s recommendations [17], we classified sarcopenia stages into (1) probable sarcopenia, which was defined by having low muscle strength; (2) sarcopenia, which was defined by having a combination of low muscle mass and low muscle strength; and (3) severe sarcopenia, which was defined by having low levels of all three components of sarcopenia. In this study, probable sarcopenia was one of the components of sarcopenia (low muscle strength), and the number of severe sarcopenia patients was too small (*n* = 65); therefore, our outcomes were sarcopenia and its three components. Details on the definition of the three components are shown in Appendix A: The definition of sarcopenia’s components.

### 2.3. Dietary Assessment

Dietary information applied for the present analyses was obtained through 24 h dietary surveys conducted using the Oxford WebQ (https://www.ceu.ox.ac.uk/research/oxford-webq, accessed on 25 March 2024) [19]. The 24 h dietary recall surveys collected participants’ intake of 206 foods and 32 beverages in the past 24 h, from which nutrient intake was calculated. Five rounds of 24 h dietary assessments were conducted from 2006 to 2012 (Appendix A). We included participants with at least one 24 h dietary assessment and calculated the mean of the repeated dietary assessments.

#### 2.3.1. Assessment of E-DII

The DII is produced based on 45 dietary components (individual foods, nutrients, and bioactive components) and is linked to 6 inflammatory biomarkers (interleukin [IL]-10, IL-6, IL-4, IL-1β, C-reactive protein, and tumor necrosis factor-α) [20]. Due to the availability of dietary nutrition data from the UK Biobank, 28 dietary components [trans-fat, *n*-6 fatty acids, *n*-3 fatty acids, total fat, protein, beta-carotene, iron, polyunsaturated fatty acids (PUFA), monounsaturated fatty acids (MUFA), cholesterol, carbohydrate, energy, selenium, riboflavin, alcohol, folate, fiber, magnesium, niacin, saturated fat, thiamine, zinc, vitamin C, vitamin E, vitamin D, vitamin A, vitamin B6, and vitamin B12] were used for E-DII calculation (Appendix A). Following the previous literature [20], there are several steps that were used to calculate the E-DII in this analysis. First, for ≥2 rounds of 24 h dietary assessments, we calculated the average intake of nutrients or foods to obtain the intake of each dietary component. Second, we adjusted energy for intake of each dietary component using the nutrient density approach [21] to control the confounding effect of energy. Third, calculating the Z score for each dietary component: energy-adjusted intake minus the mean of the database and then dividing this value by the standard deviation of the dietary component. Fourth, we converted the Z scores to percentile scores, which were then doubled and then one was further subtracted. Finally, an individual’s E-DII was obtained by multiplying the converted Z scores with the literature-derived inflammatory effect scores for specific components and then summing them together. An individual with a lower E-DII score indicates that his/her diet was more anti-inflammatory, and vice versa. We divided the E-DII into tertiles and set tertile 3 (the highest level) as the reference group.

#### 2.3.2. Assessment of DDS

Dietary diversity score (DDS), an indicator used to measure the diversity and balance of dietary intake, was developed by Kant et al. [22] and has been validated in studies in recent years [14,15]. According to a previous study [23] and the food group classification guidance of the United Nation’s Food and Agriculture Organization [24], we calculated the DDS based on five major food categories (18 subcategories) (Appendix A): grain products (nonwhole grains, whole grains), fruits (vitamin A-rich fruits, citrus and berries, and others), vegetables (vitamin A-rich vegetables, dark green leafy vegetables, starchy tubers, and others), dairy foods (cheese, milk, and yogurt), animal foods and protein alternatives (poultry, red meat, organ meat, fish and seafoods, eggs, nuts and legumes). One point was added to each DDS when participants consumed any food in each food subcategory. However, different foods consumed in the same food subcategory were not double-counted. The DDS ranged from 1 to 18 (since an individual’s reasonable diet was impossible to consume without any food in a day, we removed participants with a DDS score of 0). An individual with a higher DDS score indicates that his/her diet was richer and more balanced, and vice versa. According to practical implications for public health, we divided the DDS into three levels: low-level DDS (ranging from 1 to 6), medium-level DDS (ranging from 7 to 12), and high-level DDS (ranging from 13 to 18); low-level DDS was set as the reference group.

### 2.4. Covariates

Several covariates were considered in this analysis, which were identified based on three criteria: (1) must be related to dietary consumption; (2) must be risk factors for sarcopenia; and (3) must not be an intermediate factor between dietary consumption and sarcopenia. The covariates were assessed by questionnaires and nurse-led interviews, which included sex (females/males), age (continuous), race (nonwhite/white), household income (GBP <18,000, GBP 18,000 to 52,000, GBP >52,000), residential area (rural/urban), physical activity (PA) (hours/week) (continuous), alcohol consumption (g/week) (continuous) (Appendix A), smoking status (never/previous/current), body mass index (BMI) (kg/m^2^) (continuous), total energy intake (kcal/day) (continuous), dietary supplement (yes/no), cardiovascular disease (CVD) (yes/no), cancer (yes/no), diabetes (type 1 diabetes, type 2 diabetes, gestational diabetes, and other types of diabetes), high cholesterol (yes/no), and hypertension (yes/no). Notably, age was calculated by subtracting the date of birth from the last 24 h dietary assessment date. We used the weekly metabolic equivalent of task in hours each week to measure physical activity. The formula for calculating BMI was weight (kg)/height (meter)^2^.

### 2.5. Statistical Analysis

Continuous variables are described as the mean (standard deviation), and categorical variables were presented as the frequency (proportion). Participants’ general characteristics are shown as tertiles of E-DII and three DDS categories. We assessed the correlation between DDS and E-DII using the Spearman correlation coefficient.

Logistic regression models were used to investigate the associations of E-DII and DDS with sarcopenia and its components by calculating the odds ratios (ORs) and 95% confidence intervals (95% CIs). Based on epidemiological knowledge and previous studies, we fitted three statistical models. Model 1 was adjusted for age and sex. We further controlled for race, household income, residential area, PA, alcohol consumption, smoking status, and dietary supplements in Model 2. In Model 3 (the full model), we additionally adjusted for BMI, hyperlipemia, hypertension, diabetes, cancer, and CVD. Notably, for assessments of the effect of DDS on outcomes, we additionally adjusted for total energy intake in Models 2 and 3, and, for the three sarcopenia components (low muscle mass, low muscle strength, and low physical performance), we further mutually adjusted for three components in Models 2 and 3.

To explore the potential combined effects of E-DII and DDS on sarcopenia and its components, we first combined tertiles of E-DII with DDS levels (high, medium, and low) to create a new term with nine combinations (3 × 3) of E-DII levels and DDS levels. We further calculated ORs and their 95% CIs with the reference group of tertile 3 of E-DII and low DDS in Model 3. We also tested the multiplicative and additive interactions [25]. The multiplicative interactions were tested by including a cross-product term of E-DII and DDS, adjusted for confounders in Model 3. Specifically, we tested the multiplicative interaction effects by comparing the −2 log-likelihood that included and excluded the cross-product interaction terms of E-DII and DDS in the models. For additive interactions, we calculated the relative excess risk due to interaction (RERI) to test the presence of additive interaction [26]. RERI measures the combined excess risk in two exposed groups that are due to the interaction. RERI values equal to 0, less than 0, and more than 0 indicate no, negative, and positive additive interaction, respectively [26].

### 2.6. Sensitivity Analysis

Several sensitivity analyses were performed. First, we conducted a sensitivity analysis with 304,723 participants (Appendix A) who completed a Food Frequency Questionnaire (FFQ) assessment. The specific calculation methods are shown in the Appendix A: Assessment of DDS based on food frequency questionnaires (FFQ) in UK Biobank. Second, we defined sarcopenia as having either low muscle strength or low physical performance and low muscle mass following the EWGSOP1’s recommendations [27] and repeatedly assessed the associations of E-DII and DDS with sarcopenia. Third, to avoid potential prediction errors of bioelectrical impedance analysis at excessively high or low BMI values, we repeated our analyses after excluding 6336 excessively high or low BMI participants (≥36 or <14 kg/m^2^) (*n* = 149,333) [28]. Fourth, we repeated the main analyses by restricting participants who participated in ≥2 rounds of 24 h dietary surveys (*n* = 96,133). Finally, we reconducted the models after imputing missing data on all covariates using multiple imputation (*n* = 202,708). We assumed that data were missing at random and imputed 30 data sets [29]. We performed analysis using each imputed data set, then combined the analysis results according to Rubin’s Rule [30]. 

### 2.7. Stratified Analysis

To test the potential variations among different subgroups, stratified analyses were additionally performed by sex (males and females), age groups (≥55 and <55 years), and BMI (≥25 and <25 kg/m^2^) in Model 3.

Two-sided *p*-values smaller than 0.05 were considered statistically significant. All statistical analyses were conducted using R software (version 4.2.1).

## 3. Results

### 3.1. Participant Characteristics

For the final sample of 155,669 adults, the mean (SD) age was 57.7 (8.0) years, 81,045 (52.1%) were females, and 6325 (4.1%) were nonwhite. The range (mean) of the E-DII was −5.45 to 4.65 (0.25), and the DDS was 1 to 18 (10). Compared with participants in tertile 3 of the E-DII (the highest E-DII tertile), those in the lower E-DII groups were more likely to be male, be older, be white people, live in rural areas, be noncurrent smokers, drink more alcohol, have higher dietary supplement intake, have a lower BMI, exercise more, and consume more energy from diet (Table 1). Participants in the higher-level DDS group were more likely to be female, be older, be white people, live in rural areas, have higher family income, have a lower BMI, eat more calories, smoke fewer cigarettes, drink less alcohol, and have higher dietary supplement intake. There was no significant correlation between E-DII and DDS (the Spearman correlation coefficient was −0.27). Among the study participants (*n* = 155,669), 496 (0.3%) had sarcopenia, 8928 (5.7%) had low muscle mass, 5771 (3.7%) had low muscle strength, and 7239 (4.7%) had low physical performance (Table 1).

### 3.2. Associations of the E-DII with Sarcopenia and Its Components

We observed significant associations between the E-DII and sarcopenia and its components in three models (Table 2). For example, in Model 3, individuals in the lowest level of E-DII (tertile 1) had a lower likelihood of sarcopenia than those in the highest level (tertile 3) (OR: 0.75; 95% CI: 0.59, 0.94), with a significant dose—response relation (OR: 0.86; 95% CI: 0.78, 0.94 per SD reduction). For individual components of sarcopenia, a lower E-DII was associated with a lower likelihood of low muscle strength [(OR_tertile 1 vs. tertile 3_= 0.83, 95% CI: 0.77, 0.90); (OR: 0.92; 95% CI: 0.90, 0.94 per SD reduction)], low muscle mass [(OR_tertile 1 vs. tertile 3_= 0.80, 95% CI: 0.74, 0.87); (OR: 0.96; 95% CI: 0.93, 0.98 per SD reduction)], and low physical performance [(OR_tertile 1 vs. tertile 3_= 0.76, 95% CI: 0.71, 0.82); (OR: 0.89; 95% CI: 0.86, 0.91 per SD reduction)]. Similar associations were observed in all sensitivity analyses (Appendix A).

### 3.3. Associations of DDS with Sarcopenia and Its Components

Inverse associations were observed between DDS and the prevalence of sarcopenia and its components in three models (Table 3). For example, in Model 3, individuals in the high-level DDS group (13–18) had a lower likelihood of sarcopenia than those in the low-level DDS group (1–6) (OR: 0.69; 95% CI: 0.48, 0.95), with a significant dose—response relation (OR: 0.89; 95% CI: 0.80, 0.98 per SD increment). For individual components of sarcopenia, a higher DDS was associated with a lower likelihood of low muscle strength [(OR _high-level vs. low-level_ = 0.70, 95% CI: 0.64, 0.77); (OR: 0.89; 95% CI: 0.87, 0.92 per SD increment)], low muscle mass [(OR _high-level vs. low-level_ = 0.84, 95% CI: 0.76, 0.94); (OR: 0.94; 95% CI: 0.91, 0.97 per SD increment)], and low physical performance [(OR _high-level vs. low-level_ = 0.63, 95% CI: 0.58, 0.68); (OR: 0.85; 95% CI: 0.83, 0.87 per SD increment)]. Similar associations were observed in all sensitivity analyses (Appendix A).

### 3.4. Combined Effects of the E-DII and DDS on Sarcopenia and Its Components

When we investigated the association of combined categories of E-DII and DDS, within each tertile of E-DII, higher-level DDS was associated with a lower OR of sarcopenia and its three components (Table 4). Compared with individuals who were in the group with low-level DDS and tertile 3 of E-DII, those in the group with high-level DDS and tertile 1 of E-DII had the lowest prevalence of sarcopenia (OR: 0.69; 95% CI: 0.65, 0.74), low muscle strength (OR: 0.62; 95% CI: 0.55, 0.70), low muscle mass (OR: 0.74; 95% CI: 0.65, 0.85), and low physical performance (OR: 0.54; 95% CI: 0.48, 0.61).

We observed a clear multiplicative interaction between DDS and E-DII for sarcopenia (*p*-interaction < 0.001) and low physical performance (*p*-interaction = 0.003) in Model 3. However, there was no multiplicative interaction (*p*-interaction > 0.05) for low muscle mass and low muscle strength (Table 4). The test for additive interactions was not significant for sarcopenia and its components (all 95% CIs of RERIs covered 0) (Table 5). Similar associations were observed in all sensitivity analyses (Appendix A).

### 3.5. Stratified Analysis

Similar associations were observed in all stratified analyses (Appendix A), which showed that the association of E-DII or DDS with the risk of sarcopenia and its components may not be modified by age, sex, and BMI (*p*-interaction > 0.05).

## 4. Discussion

In this cross-sectional study of the UK Biobank, lower E-DII and higher DDS were associated with a lower prevalence of sarcopenia and its three components (low muscle mass, low muscle strength, and low physical performance). Furthermore, we observed significant combined associations between E-DII and DDS with sarcopenia and low physical performance on the multiplicative interaction scale. Our findings may provide the following two insights into the prevention of sarcopenia: first, anti-inflammatory diet and dietary diversity play pivotal roles in preventing the occurrence of sarcopenia and its three components; second, more cases of sarcopenia and low physical performance might be preventable by adherence to a more anti-inflammatory diet combined with a higher dietary diversity.

### 4.1. Comparison with Other Studies and Possible Explanations

In this study, fewer sarcopenia events were identified according to EWGSP2’s recommendations in comparison with EWGOP1’s recommendations (the prevalence was 0.32% versus 9.12%). In accordance with our results, previous studies also found a lower prevalence estimate using the EWGSP2 guideline compared with the EWGOP1 guideline [31,32]. However, our findings show a relatively lower prevalence (0.32%) applying the new EWGSP2 guideline. There are two possible reasons for this difference: one is that participants in the UK Biobank are relatively healthy individuals in terms of socioeconomic status and lifestyle, and the other is that the UK Biobank participants are overall younger (the average age of UK Biobank participants is 55.7 years) than several other study samples. In other words, the prevalence of sarcopenia applying the new EWGSP2 guideline could be lower in a younger population. However, it is worth noting that, under the new criteria, for sarcopenia definition, there is a subset of people who are not identified as having sarcopenia but are still a high-risk group. Therefore, while focusing on events with sarcopenia, attention should also be paid to those with the three components of sarcopenia that occur.

A significant positive association between the E-DII and sarcopenia was observed, and this association remained significant between the E-DII and the three components of sarcopenia. In accordance with our findings, the association between DII and the prevalence of sarcopenia was also observed in three previous studies [9,10,11]. For example, a cross-sectional analysis conducted among 140 Crohn’s disease patients from Ruijin Hospital in Shanghai showed that a higher DII score significantly increased the risk of sarcopenia [9]. Jiwen Geng et al. reported that the risk of sarcopenia increased with increasing DII tertiles [11]. However, controversial evidence remains. For instance, in a cross-sectional analysis conducted among 300 adults aged ≥55 years, no significant association between DII and three components of sarcopenia (abnormal handgrip, low muscle mass, and abnormal gait speed) was observed [10]. Differences in dietary measurements may explain the heterogeneity to some extent: we used dietary data from multiple 24 h dietary assessments, while Amir Bagheri et al. used dietary information from one FFQ survey. Apart from the heterogeneity in population, sample size and study period, the reasons behind the different findings require further investigation. The mechanisms linking the positive association of the E-DII with sarcopenia may be related to anti-inflammatory biomarkers. An anti-inflammatory diet is linked to anti-inflammatory biomarkers (interleukin [IL]-10, IL-6, IL-4, IL-1β, C-reactive protein, and tumor necrosis factor-α) [20]. Previous studies have indicated that these inflammatory biomarkers inversely correlate with muscle strength [33] and that C-reactive protein levels inversely correlate with sarcopenia [34]. Moreover, a systematic review suggested that anti-inflammatory therapy might reduce inflammation-induced muscle weakness [35].

We found that DDS was negatively associated with the ORs of sarcopenia and its components. No study has evaluated DDS in relation to sarcopenia, while one cross-sectional study revealed that deficient dietary diversity, which was defined as hardly ever eating any of 10 foods (potatoes, soybean products, seaweeds, fruits, green and yellow vegetables, meat, milk, eggs, fish/shellfish, and oil and fats) in the past week, was associated with sarcopenia in older participants [36]. Moreover, several studies have investigated other healthy dietary patterns associated with sarcopenia. For example, Mikael Karlsson et al. found that a Mediterranean-like diet and a healthy diet indicator are more likely to protect against the occurrence of sarcopenia [37], and C.H. Huang et al. reported that adherence to the Japanese Food Guide Spinning Top potentially improved muscle mass in community-dwelling older adults. The DDS with these healthy dietary patterns is characterized by greater intake of whole grains, vegetables, fruits, seeds and nuts, and seafoods, and can be used to quantify an individual’s overall diet quality [38]. In addition, as recently reviewed, diet quality defined by diversity and nutrient adequacy was negatively associated with low physical performance, low muscle strength, and low muscle mass [39]. The mechanisms underlying the observed negative associations of DDS with sarcopenia and its components remain unclear. However, previous studies have provided some indication of the benefits of higher intake of vitamin D and B vitamins [40], magnesium [41], and antioxidant nutrients [42] for muscle health.

A novel finding of this study was the joint effects of E-DII and DDS on sarcopenia and low physical performance. This finding suggests that a combination of a lower Dietary Inflammatory Index (E-DII) and a higher Dietary Diversity Score (DDS) is more favorably associated with reduced prevalence of sarcopenia and low physical performance, compared to the impact of a lower E-DII or a higher DDS in isolation. Furthermore, at the same level of E-DII, the risk of individuals who adhered to a higher DDS was lower than that of those with a lower DDS. However, within the same DDS subgroup, participants adhering to a more anti-inflammatory diet did not have a significantly lower prevalence than those with a less anti-inflammatory diet. Our results also suggest that anti-inflammatory diets are important to prevent the development of sarcopenia, especially when dietary intake is sufficiently varied. The potential mechanisms have yet to be clarified. The plausible biological effects of these two factors likely involve overlapping influences on sarcopenia and low physical performance. One of the reasons behind this finding may be that both anti-inflammatory and high-diversity diets share common dietary elements: higher whole grains, vegetables, fruits, high-quality protein and fat, dietary fiber, and nut intake. Consumption of these healthy foods could impact sarcopenia-related mechanisms, such as anti-inflammation, antioxidative stress, maintenance of high-quality protein, enhancement of immune function, and stabilization of hormonal levels [16].

### 4.2. Strengths and Limitations

The major strength was the novelty of our analyses; to our knowledge, ours is the first study to assess the joint effect of dietary inflammatory potential and dietary diversity on sarcopenia. Other strengths included using the new EWGSOP2 definition, the detailed and repeated assessments of dietary information, and the large sample population.

Several limitations should also be mentioned. First, with collecting data related to diet and sarcopenia simultaneously as a snapshot, cross-sectional data cannot be used to study the temporal order and determine a causal relationship. Second, dietary information was self-reported, which may involve recall bias. However, five rounds of 24 h dietary assessments were applied, reducing possible measurement errors. Additionally, DDS was calculated using FFQ, and the analyses were repeated to test the reliability of dietary information. Third, participants in the UK Biobank do not fully represent the UK general population regarding socioeconomic status and lifestyle. Fourth, owing to the low prevalence of sarcopenia applying the new EWGSOP2 definition, statistical power may be insufficient to detect significant associations between the E-DII (or DDS) and sarcopenia. Finally, although we adjusted key covariates based on epidemiology knowledge and previous studies, residual confounding remained possible, and causality cannot be determined because of shortcomings in observational studies.

## 5. Conclusions

Our study suggests that both lower dietary inflammatory potential and higher dietary diversity might play important roles in preventing the occurrence of sarcopenia and its three components. It might be possible to prevent more cases of sarcopenia and low physical performance by adhering to an anti-inflammatory diet that also boasts high dietary diversity, according to associations observed in our cross-sectional study. However, further research, particularly longitudinal or intervention studies, is needed to establish a causal relationship.

## Figures and Tables

**Table 1 nutrients-16-01038-t001:** General characteristics of participants from UK Biobank according to E-DII and DDS subgroups ^a^.

Variables	Overall	E-DII			DDS		
		Tertile 1 (−5.45, −0.49)	Tertile 2 (−0.50, 1.12)	Tertile 3 (1.13, 4.65)	Low (1–6)	Medium (7–12)	High (13–18)
No. of participants	155,669	51,890	51,889	51,890	18,838	95,428	41,403
Age (years), mean (SD)	57.7 (8.0)	58.4 (7.9)	57.8 (7.9)	56.9 (8.0)	55.7 (8.1)	57.6 (8.0)	58.9 (7.6)
Sex, female	81,045 (52.1)	23,498 (45.3)	27,527 (53.0)	30,020 (57.9)	8291 (44.0)	48,819 (51.2)	23,935 (57.8)
Race, nonwhite	6325 (4.1)	1766 (3.4)	1733 (3.3)	2826 (5.4)	1295 (6.9)	3869 (4.1)	1161 (2.8)
Residence, rural	24,973 (16.0)	8554 (16.5)	8449 (16.3)	7970 (15.4)	2616 (13.9)	15,346 (16.1)	7011 (16.9)
Household income							
GBP >52,000	51,704 (33.2)	16,706 (32.2)	17,852 (34.4)	17,146 (33.0)	5746 (30.5)	31,584 (33.1)	14,374 (34.7)
GBP 18,000 to 52,000	81,437 (52.3)	27,538 (53.1)	27,047 (52.1)	26,852 (51.7)	9672 (51.3)	49,928 (52.3)	21,837 (52.7)
GBP <18,000	22,528 (14.5)	7646 (14.7)	6990 (13.5)	7892 (15.2)	3420 (18.2)	13,916 (14.6)	5192 (12.5)
Smoking status							
Never	87,650 (56.3)	29,374 (56.6)	29,513 (56.9)	28,763 (55.4)	9879 (52.4)	53,554 (56.1)	24,217 (58.5)
Previous	55,824 (35.9)	19,178 (37.0)	18,712 (36.1)	17,934 (34.6)	6366 (33.8)	34,371 (36.0)	15,087 (36.4)
Current	12,195 (7.8)	3338 (6.4)	3664 (7.1)	5193 (10.0)	2593 (13.8)	7503 (7.9)	2099 (5.1)
Alcohol consumption (g/week), mean (SD)	114.9 (99.5)	119.6 (102.8)	114.7 (98.6)	110.4 (96.6)	131.7 (121.9)	115.6 (99.5)	105.7 (85.9)
PA, MET (hours/week), mean (SD)	41.1 (40.1)	45.2 (42.5)	39.8 (39.0)	38.2 (39.2)	41.8 (44.5)	40.9 (40.4)	41.2 (38.3)
BMI (kg/m^2^), mean (SD)	26.9 (4.5)	26.7 (4.5)	26.8 (4.4)	27.1 (4.7)	27.7 (4.8)	26.9 (4.5)	26.5 (4.4)
Energy intake (kcal/day), mean (SD)	2087.8 (617.7)	2465.5 (598.7)	2085.9 (516.3)	1711.9 (485.4)	1884.6 (636.5)	2091.0 (612.6)	2172.8 (599.5)
Dietary supplement	73,379 (47.1)	26,216 (50.5)	24,418 (47.1)	22,745 (43.8)	7794 (41.4)	44,446 (46.6)	21,139 (51.1)
Diabetes	6345 (4.1)	2208 (4.3)	1974 (3.8)	2163 (4.2)	905 (4.8)	3945 (4.1)	1495 (3.6)
CVD	6348 (4.1)	2251 (4.3)	2013 (3.9)	2084 (4.0)	905 (4.8)	3908 (4.1)	1535 (3.7)
Cancer	13,262 (8.5)	4536 (8.7)	4342 (8.4)	4384 (8.4)	1433 (7.6)	8005 (8.4)	3824 (9.2)
Hypertension	34,983 (22.5)	12,001 (23.1)	11,634 (22.4)	11,348 (21.9)	4390 (23.3)	21,209 (22.2)	9384 (22.7)
Hyperlipidemia	16,207 (10.4)	5669 (10.9)	5325 (10.3)	5213 (10.0)	1953 (10.4)	9878 (10.4)	4376 (10.6)
Sarcopenia	496 (0.3)	156 (0.3)	146 (0.3)	194 (0.4)	57 (0.3)	308 (0.3)	131 (0.3)
Low muscle mass	8928 (5.7)	2793 (5.4)	3023 (5.8)	3112 (6.0)	851 (4.5)	5368 (5.6)	2709 (6.5)
Low muscle strength	5771 (3.7)	1799 (3.5)	1854 (3.6)	2118 (4.1)	840 (4.5)	3483 (3.6)	1448 (3.5)
Low physical performance	7239 (4.7)	2094 (4.0)	2122 (4.1)	3023 (5.8)	1337 (7.1)	4399 (4.6)	1503 (3.6)

^a^ Values are numbers (percentages) unless stated otherwise. Abbreviations: BMI, body mass index; CVD, cardiovascular disease; DDS, dietary diversity score; E-DII, dietary inflammatory index; MET, metabolic equivalent; PA, physical activity; SD, standard deviation.

**Table 2 nutrients-16-01038-t002:** Associations of E-DII with sarcopenia and its components in UK Biobank (*n* = 155,669).

	E-DII, ORs (95% Cls)			Continuous (per SD Reduction)
	Tertile 1 (−5.45, −0.49)	Tertile 2 (−0.50, 1.12)	Tertile 3 (1.13, 4.65)	
No. of participants	41,403	95,428	18,838	155,669
Sarcopenia				
Cases	156	146	194	496
Model 1 ^a^	0.62 (0.47, 0.80)	0.67 (0.53, 0.84)	1.00 (Ref)	0.83 (0.76, 0.92)
Model 2 ^a^	0.71 (0.54, 0.91)	0.71 (0.51, 0.97)	1.00 (Ref)	0.84 (0.77, 0.93)
Model 3 ^a^	0.75 (0.59, 0.94)	0.76 (0.59, 0.98)	1.00 (Ref)	0.86 (0.78, 0.94)
Low muscle strength				
Cases	1799	1854	2118	5771
Model 1 ^b^	0.79 (0.74, 0.84)	0.83 (0.78, 0.89)	1.00 (Ref)	0.89 (0.87, 0.92)
Model 2 ^b^	0.84 (0.82, 0.94)	0.88 (0.78, 0.90)	1.00 (Ref)	0.92 (0.90, 0.94)
Model 3 ^b^	0.83 (0.77, 0.90)	0.88 (0.82, 0.94)	1.00 (Ref)	0.92 (0.90, 0.94)
Low muscle mass				
Cases	2793	3023	3112	8928
Model 1 ^b^	0.77 (0.72, 0.83)	0.79 (0.74, 0.84)	1.00 (Ref)	0.93 (0.90, 0.95)
Model 2 ^b^	0.78 (0.72, 0.85)	0.80 (0.75, 0.86)	1.00 (Ref)	0.93 (0.91, 0.96)
Model 3 ^b^	0.80 (0.74, 0.87)	0.88 (0.92, 0.95)	1.00 (Ref)	0.96 (0.93, 0.98)
Low physical performance				
Cases	2094	2122	3023	7239
Model 1 ^b^	0.65 (0.61, 0.69)	0.67 (0.63, 0.71)	1.00 (Ref)	0.81 (0.79, 0.83)
Model 2 ^b^	0.73 (0.68, 0.78)	0.75 (0.71, 0.81)	1.00 (Ref)	0.88 (0.86, 0.91)
Model 3 ^b^	0.76 (0.71, 0.82)	0.76 (0.72, 0.81)	1.00 (Ref)	0.89 (0.86, 0.91)

^a^ ORs (95% CIs) of E-DII with sarcopenia were examined using logistic regression models; Model 1 was adjusted for age and sex; Model 2 additionally included race, household income, residence, smoking status, alcohol consumption, physical activity, and dietary supplement; Model 3 additionally included BMI, diabetes, CVD, cancer, hypertension, and hyperlipidemia. ^b^ ORs (95% CIs) of E-DII with sarcopenia’s components were examined using logistic regression models; Model 1 was adjusted for age and sex; Model 2 additionally included race, household income, residence, smoking status, alcohol consumption, physical activity, dietary supplement, and three components (low muscle strength, low muscle mass, and low physical performance) were mutually adjusted; Model 3 additionally included BMI, CVD, diabetes, cancer, hypertension, and hyperlipidemia. Abbreviations: BMI, body mass index; CIs: confidence intervals; CVD, cardiovascular disease; E-DII, dietary inflammatory index; ORs: odds ratios; Ref, reference; SD, standard deviation.

**Table 3 nutrients-16-01038-t003:** Associations of DDS with sarcopenia and its components in UK Biobank (*n* = 155,669).

	ORs (95% Cls)			Continuous (per SD Increment)
	Low-Level DDS (1–6)	Medium-Level DDS (7–12)	High-Level DDS (13–18)	
No. of participants	25,255	123,362	52,311	155,669
Sarcopenia				
Cases	57	308	131	496
Model 1 ^a^	1.00 (Ref)	0.75 (0.55, 1.05)	0.62 (0.43, 0.90)	0.87 (0.79, 0.95)
Model 2 ^a^	1.00 (Ref)	0.80 (0.56, 1.05)	0.70 (0.47, 0.99)	0.87 (0.79, 0.97)
Model 3 ^a^	1.00 (Ref)	0.84 (0.62, 1.14)	0.69 (0.48, 0.95)	0.89 (0.80, 0.98)
Low muscle strength				
Cases	840	3483	1448	5771
Model 1 ^b^	1.00 (Ref)	0.70 (0.65, 0.76)	0.61 (0.55, 0.66)	0.85 (0.83, 0.88)
Model 2 ^b^	1.00 (Ref)	0.76 (0.70, 0.82)	0.69 (0.63, 0.76)	0.88 (0.86, 0.91)
Model 3 ^b^	1.00 (Ref)	0.77 (0.71, 0.83)	0.70 (0.64, 0.77)	0.89 (0.87, 0.92)
Low muscle mass				
Cases	851	5368	2709	8928
Model 1 ^b^	1.00 (Ref)	0.80 (0.72, 0.79)	0.76 (0.67, 0.86)	0.91 (0.89, 0.93)
Model 2 ^b^	1.00 (Ref)	0.83 (0.75, 0.93)	0.81 (0.72, 0.88)	0.93 (0.90, 0.95)
Model 3 ^b^	1.00 (Ref)	0.88 (0.80, 0.97)	0.84 (0.76, 0.94)	0.94 (0.91, 0.97)
Low physical performance				
Cases	1337	4399	1503	7239
Model 1 ^b^	1.00 (Ref)	0.57 (0.54, 0.61)	0.42 (0.38, 0.45)	0.74 (0.72, 0.76)
Model 2 ^b^	1.00 (Ref)	0.67 (0.63, 0.72)	0.55 (0.51, 0.60)	0.80 (0.78, 0.82)
Model 3 ^b^	1.00 (Ref)	0.74 (0.69, 0.79)	0.63 (0.58, 0.68)	0.85 (0.83, 0.87)

^a^ ORs (95% CIs) of DDS with sarcopenia were examined using logistic regression models; Model 1 was adjusted for age and sex; Model 2 additionally included race, household income, residence, smoking status, alcohol consumption, physical activity, dietary supplement, and total calorie intake from diet; Model 3 additionally included BMI, diabetes, CVD, cancer, hypertension, and hyperlipidemia. ^b^ ORs (95% CIs) of DDS with sarcopenia’s components were examined using logistic regression models; Model 1 was adjusted for age and sex; Model 2 additionally included race, household income, residence, smoking status, alcohol consumption, physical activity, dietary supplement, total calorie intake from diet, and three components (low muscle strength, low muscle mass, and low physical performance) were mutually adjusted; Model 3 additionally included BMI, diabetes, CVD, cancer, hypertension, and hyperlipidemia. Abbreviations: BMI, body mass index; CIs: confidence intervals; CVD, cardiovascular disease; DDS, dietary diversity score; ORs: odds ratios; Ref, reference; SD, standard deviation.

**Table 4 nutrients-16-01038-t004:** Combined associations of E-DII and DDS with sarcopenia and its components in the UK Biobank (*n* = 155,669).

	ORs (95% CIs) ^c^			*p*-Interaction ^d^
	Tertile 3 of E-DII ^b^	Tertile 2 of E-DII ^b^	Tertile 1 of E-DII ^b^	
Sarcopenia				<0.001
Low-level DDS ^a^	1.00 (Ref)	0.90 (0.82, 0.99)	0.97 (0.84, 1.12)	
Medium-level DDS ^a^	0.83 (0.79, 0.87)	0.76 (0.72, 0.80)	0.75 (0.71, 0.79)	
High-level DDS ^a^	0.79 (0.65, 0.85)	0.71 (0.67, 0.75)	0.69 (0.65, 0.74)	
Low muscle strength				0.265
Low-level DDS ^a^	1.00 (Ref)	0.83 (0.69, 1.00)	1.03 (0.78, 1.35)	
Medium-level DDS ^a^	0.77 (0.70, 0.86)	0.72 (0.65, 0.79)	0.72 (0.64, 0.80)	
High-level DDS ^a^	0.74 (0.64, 0.86)	0.71 (0.63, 0.80)	0.62 (0.55, 0.70)	
Low muscle mass				0.884
Low-level DDS ^a^	1.00 (Ref)	1.02 (0.80, 1.28)	0.81 (0.55, 1.18)	
Medium-level DDS ^a^	0.92 (0.83, 1.02)	0.88 (0.78, 0.98)	0.75 (0.66, 0.86)	
High-level DDS ^a^	0.87 (0.66, 1.00)	0.85 (0.74, 0.97)	0.74 (0.65, 0.85)	
Low physical performance				0.003
Low-level DDS ^a^	1.00 (Ref)	0.90 (0.77, 1.05)	0.83 (0.64, 1.06)	
Medium-level DDS ^a^	0.80 (0.74, 0.88)	0.65 (0.59, 0.72)	0.65 (0.59, 0.71)	
High-level DDS ^a^	0.68 (0.58, 0.78)	0.61 (0.54, 0.68)	0.54 (0.48, 0.61)	

^a^ DDS levels [low (1–6), medium (7–12), and high (13–18)] were defined according to practical implications for public health. ^b^ Tertile 1 of E-DII ranged from −5.44 to −0.49; tertile 2 of E-DII ranged from −0.50 to 1.12 and from 1.13 to 4.65 in UK Biobank. ^c^ Combined associations of E-DII and DDS with sarcopenia were assessed based on covariates in Model 3: age, sex, race, household income, residence, smoking status, alcohol consumption, physical activity, dietary supplement, BMI, diabetes, CVD, cancer, hypertension, and hyperlipidemia; three components (low muscle strength, low muscle mass, and low physical performance) were additionally mutually adjusted in Model 3 while assessing the combined associations of E-DII and DDS with sarcopenia’ components. ^d^ We tested the multiplicative interaction effects by comparing the −2 log-likelihood that included and excluded the cross-product interaction terms of E-DII and DDS in Model 3. Abbreviations: BMI, body mass index; CIs: confidence intervals; CVD, cardiovascular disease; DDS, dietary diversity score; E-DII, dietary inflammatory index; ORs: odds ratio; Ref, references.

**Table 5 nutrients-16-01038-t005:** RERI of E-DII and DDS on the risk of sarcopenia and its components in the UK Biobank (*n* = 155,669).

	RERI (95% CIs) ^c^		
	Tertile 3 of E-DII ^b^	Tertile 2 of E-DII ^b^	Tertile 1 of E-DII ^b^
Sarcopenia			
Low-level DDS ^a^	1.00 (Ref)	-	-
Medium-level DDS ^a^	-	0.04 (−0.04, 0.11)	−0.01 (−0.11, 0.08)
High-level DDS ^a^	-	−0.08 (−0.37, 0.21)	−0.09 (−0.43, 0.18)
Low muscle strength			
Low-level DDS ^a^	1.00 (Ref)	-	-
Medium-level DDS ^a^	-	−0.06 (−0.20, 0.08)	−0.08 (−0.25, 0.09)
High-level DDS ^a^	-	−0.27 (−0.67, 0.13)	−0.10 (−0.48, 0.28)
Low muscle mass			
Low-level DDS ^a^	1.00 (Ref)	-	-
Medium-level DDS ^a^	-	0.02 (−0.11, 0.16)	0.06 (−0.10, 0.22)
High-level DDS ^a^	-	−0.09 (−0.52, 0.34)	−0.03 (−0.43, 0.37)
Low physical performance			
Low-level DDS ^a^	1.00 (Ref)	-	-
Medium-level DDS ^a^	-	0.10 (−0.02, 0.22)	0.03 (−0.13, 0.20)
High-level DDS ^a^	-	0.09 (−0.25, 0.42)	−0.05 (−0.38, 0.28)

^a^ DDS levels [low (1–6), medium (7–12), and high (13–18)] were defined according to practical implications for public health. ^b^ Tertile 1 of E-DII ranged from −5.44 to −0.49; tertile 2 of E-DII ranged from −0.50 to 1.12 and from 1.13 to 4.65 in UK Biobank. ^c^ RERI (95% CIs) values were calculated based on the reference group with tertile 3 of E-DII and low-level DDS. RERI values equal to 0, less than 0, and more than 0 indicate no, negative, and positive additive interaction, respectively. Abbreviations: CIs: confidence intervals; DDS, dietary diversity score; E-DII, dietary inflammatory index; RERI, the relative excess risk due to interaction.

## Data Availability

The datasets generated and analyzed during the current study are available upon reasonable request to the Access Management System (AMS) through the UK Biobank website (https://www.ukbiobank.ac.uk/enable-your-research/apply-for-access, accessed on 25 March 2024).

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
