# Peer review of "Dietary Inflammatory Index and Dietary Diversity Score Associated with Sarcopenia and Its Components: Findings from a Nationwide Cross-Sectional Study"

_nutrients, 2024, doi:10.3390/nu16071038_

Round 1
Reviewer 1 Report
Comments and Suggestions for Authors
This is important work, with scientifically sound statistical analyses plan. However, there are many issues that need to be addressed i.e., the dietary analyses.
Abstract:
- what does one (maximum) mean? do you mean ranging 1 to 5?
Methods
- how did you determine inplausible faily intake? 3500kcal is possible for females. Please reference.
For diabetes, are you refering to both type 1 and 2? if only type 2, please use the appropriate term.
- With the age group for stratified analyses, why was 55 years instead of 65 chosen?
- please report how the data was retrieved and its availability
- The formula for calculating BMI was weight (kg)/height2. - Metres?
Results
- the alcohol consumed seem a little much at 10 standard drinks a week, which means 2 a day for 5 days. The results does not tally which the mean caloric intake being ~2000, alcohol contributing to ?40% of their energy intake? Same goes for the MET for physical activity, please check.
- more transparency needed to demonstrate how E-DII is calculated from the diet. Was it from the readily from the biobank ?
Discussion and overall, please change the following:
- Elderly to older adults
- once an acronym has been described, please ensure that it is used throughout the manuscript i.e., no need to relist them again in latter parts of the manuscript.
- extreme BMIs need to be reworded in sensitivity analyses.
Author contributions does not meet guidelines.
Comments on the Quality of English Languageacceptable.
Reviewer 2 Report
Comments and Suggestions for Authors
Abstract section: According to instructions for authors “A single paragraph of about 200 words maximum” and your abstract has 232 words.
The number of people in this study was 211,025 participants, which demonstrated a good methodology
In 2.4 Covariates, you should change this (<18,000£, 18,000 to 51,999£, ≥52,000£) for (<18,000£, 18,000 to 52,000£, >52,000£).
In line 162, you should add the units of height due to that weight is reflected in kg.
Discussion and Conclusions sections is very short. Please you should to increase these sections.
Round 2
Reviewer 1 Report
Comments and Suggestions for Authors
The authors have answered my queries, no further questions.
Author Response
Thanks again for taking your time to review this manuscript.